# Low-Cost UWB Based Real-Time Locating System: Development, Lab Test, Industrial Implementation and Economic Assessment

**DOI:** 10.3390/s23031124

**Published:** 2023-01-18

**Authors:** Andrea Volpi, Letizia Tebaldi, Guido Matrella, Roberto Montanari, Eleonora Bottani

**Affiliations:** Department of Engineering and Architecture, University of Parma, Parco Area delle Scienze 181/A, 43124 Parma, Italy

**Keywords:** Real-Time Locating System, Ultra-Wideband, sensor network, assets tracking, warehouse management, wireless communications, KPI

## Abstract

This paper presents the technical development and subsequent testing of a Real-Time Locating System based on Ultra-Wideband signals, with the aim to appraise its potential implementation in a real industrial case. The system relies on a commercial Radio Indoor Positioning System, called Qorvo MDEK1001, which makes use of UWB RF technology to determine the position of RF-tags placed on an item of interest, which in turn is located in an area covered by specific fixed antennas (anchors). Testing sessions were carried out both in an Italian laboratory and in a real industrial environment, to determine the best configurations according to some selected performance indicators. The results support the adoption of the proposed solution in industrial environments to track assets and work in progress. Moreover, most importantly, the solution developed is cheap in nature: indeed, normally tracking solutions involve a huge investment, quite often not affordable above all by small-, medium- and micro-sized enterprises. The proposed low-cost solution instead, as demonstrated by the economic assessment completing the work, justifies the feasibility of the investment. Hence, results of this paper ultimately constitute a guidance for those practitioners who intend to adopt a similar system in their business.

## 1. Introduction

With technologies available for wireless communication improving, wireless networks (WNs) have gained increasing interest from academic communities and private companies and have shown application potentials in various fields [1]. The reasons for this success are to be found in the main advantages brought by WNs, including flexibility, low cost and ease of deployment, which make the technology promising and vital in the Industry 4.0 framework. Smart factories and intelligent manufacturing systems can benefit from the deployment of industrial wireless networks (IWNs) as the vital foundations for realizing the architecture of Industry 4.0 and smart factories [2].

In recent years, wireless sensor networks (WSNs) have been addressed by researchers and industry, leading to a large number of applications of WSNs in agriculture, military, health and other domains [3]. A WSN consists of a large number of ordinary nodes whose main function is sensing physical parameters, such as temperature, humidity, voice and other parameters of the monitored environment or device [4]. Generally, WSN nodes are stationary, have limited power and do not consider the industrial environment and other special requirements, such as reliability, latency and flexibility. As interest grows, mobile nodes have been introduced into industrial systems [5]; in particular, radio modules have been mounted on mobile devices to increase flexibility and mobility, although not encompassed by traditional WSNs.

IWNs and WSNs share several common features, such as communication protocols and industrial applications; nonetheless, there are also some important differences, namely: latency (lower for IWNs to sense real time parameters, higher for WSNs to save energy); mobility (moving nodes in IWNs, stationery in WSNs); environment (rough industrial for IWNs, more friendly for WSNs); capacity (higher for IWNs nodes to handle data processing, energy and storage and lower for WSNs nodes) [6].

WSNs in industrial applications can be used to provide an example for IWNs: more precisely, WSNs can be deployed in smart plants, industrial environment monitoring and automation factories for low latency wireless communication [7]. In detail, determining the position of assets in industrial environments is a key activity of production plants; for instance, when dealing with job shops [8] or manually assembly lines [9], or even for managing warehouses [10], in which a large number of items have to be simultaneously tracked with high spatial accuracy. Localization of nodes in a WSN is one of the core functionalities required by many applications [11]; the peculiarity of these cases is that the positioning has to be performed indoor, namely inside buildings; hence, traditional systems such as the renowned Global Positioning System (GPS) are not suitable, since satellite radio signals cannot penetrate solid walls and obstacles [12]. This fact generated another branch of research, referred as Indoor Positioning System (IPS), extremely challenging since these environments require higher precision and accuracy, have greater density of obstacles that cause attenuation and signal scattering and are contaminated by the presence of moving people that modify the propagation channel [13].

According to [14], the most widely available wireless technologies for indoor location purposes are Wifi, Bluetooth, Zigbee, Radio Frequency Identification (RFID), Ultrasound, iBeacons and Ultra-wideband (UWB), and there is a plethora of studies dealing with their development, applications and implementations at the industrial level. Among the solutions for indoor positioning, the use of UWB ranging is widely deployed when there is the need for ensuring very high accuracy in position determination [15]. Indeed, their large bandwidth guarantees an accurate estimate by reducing the impact of errors due to signal reflections caused by walls and obstacles. Moreover, after the definition of the IEEE 802.15.4a WSN standard [16], the interest in UWB-WSN systems for indoor RTLS grew fast.

Before mentioning some interesting works, however, it is necessary to highlight that sometimes the term IPS is referred to as Real-Time Location System (RTLS), as it is a real-time technology; in this paper as well, we will use the term RTLS to further stress the real-time ability to track objects of the system in question (in other words, implicitly IPSs are also RTLs).

Going back to some existing examples within the industrial context, it is worth mentioning the application by [17], who proposed an RTLS based on RFID and UWB for digital manufacturing workshops and performed a simulated application in an experimental environment [10], instead, applied the Bluetooth technology for tracking assets in a warehouse. Again, in the field of warehouse management, Ref. [18] deployed an UWB solution for enhancing safety and operational efficiency, while recently, ref. [19] discussed the development of a standard for tracking systems in production and logistics. Under a more general perspective, Ref. [20] defined the state-of-art of RTLSs for asset tracking in Smart Manufacturing, demonstrating their usefulness for bottlenecks identification, factory layouts optimization, more informed planning and scheduling and overall improvement of productivity. In the context of production management, Ref. [21] reviewed the possible technologies and applications related to RTLS for production control and logistics, quality management, safety and efficiency monitoring in manufacturing contexts; Ref. [22] as well analyzed the existing IPSs, but from a more technical point of view (on the basis of different location predication approaches); in their study, Ref. [23] defined RTLS as a Lean Management tool for productivity improvement. Several other examples could be mentioned demonstrating the relevance of such systems, and at the same time, their huge cost. Indeed, this is one of the main issues that companies have to face when pondering the implementation of this kind of solution, and quite often results in a surrender primarily for smaller enterprises for economic reasons. This is the premise behind the system presented in this paper: Indeed, the ultimate aim is to provide a low-cost solution, at the same time effective, reliable and accurate, suitable for those micro-small sized manufacturers with limited investment capability, who intend to implement an RTLS. According to the review carried out by [13] about the main technologies for solving indoor localization issues, to be honest, RFID is the one that better satisfies the affordability in economic terms (as also confirmed by [24]), but at the same time, UWB is recognized as being the best one in terms of efficiency [18]. This statement justifies the choice of the technology implemented in this research, which relies on UWB.

Indeed, in the present manuscript, the proposed RTLS is based on a commercial Radio IPS, called Qorvo MDEK1001 (https://www.qorvo.com/, accessed on 12 December 2022), which uses UWB RF technology to determine the position of RF-tags placed on the item of interest, which is located in an area covered by specific fixed antennas, called “anchors”. Technical development, laboratory tests in pure open space conditions and in situ validation in a real industrial environment (specifically, a production warehouse) are performed, to provide a complete implementation path and performances assessment. This part of the research was carried out in partnership with an Italian company based in Parma, in the north of Italy, anonymous for the sake of privacy.

Other previous studies specifically focus on the Qorvo MDEK1001, but before mentioning them, it is important to clarify that two years ago, the Qorvo brand acquired the Decawave brand, and according to this fact, until 2020, the device in question was named Decawave MDEK1001. In the light of this fact, no previous scientific studies refer to “Qorvo MDEK1001”; actually, they refer to the previous name, i.e., Decawave MDEK1001, although describing the same device as that used in this study.

The studies in question, for instance, have been by [25], who studied how to bypass one of the limits of Decawave MDEK1001′s protocol, namely the fact that each tag to be located can only measure ranges to a maximum of four anchors, and they carried out tests in laboratories and in an apartment, but not in a real industrial context; again, Ref. [26] designed and manufactured a mechanically flexible textile antenna-backed sensor node by applying Qorvo MDEK1001, again without any real application; Ref. [27] deepened the topic of self-calibration for the time difference of arrival positioning when dealing with the system in question; Ref. [28] dealt with its clock drift and signal power, from a technical perspective, while ref. [29] analyzed the topic of scalability for high user densities.

To the best of the knowledge of the authors, however, existing studies mainly deal with technical issues of RTLSs; as far as tests are concerned, if any, they are limited at the laboratory level. According to that, it goes without saying the contribution of the present manuscript: in addition to some technical details and laboratory experiments, here the implementation in a real warehouse is proposed, including all the steps for application. In other words, it can be considered a real development path which includes guidelines on how to set up a working RTLS system with reasonable positioning accuracy, which may be implemented in different realities and above all, as already emphasized, in those micro-small sized companies which intend to implement a RLTS in their business at an affordable investment. Supporting this last economic aspect, a comparative analysis was also performed, aimed at assessing the total cost of ownership of the Qorvo system in question compared to other common RTLSs provided by some major brands of the automation field, demonstrating the economic suitability.

The remainder of the paper is structured as follows: Section 2 illustrates the methodology followed for the laboratory test, whose results are proposed in Section 3. The two following sections, i.e., Section 4 and Section 5, recall the same structure of Section 2 and Section 3, but they are dedicated to the in situ application. Section 6 proposes a brief discussion on the achieved results, followed by Section 7, presenting the economic assessment. Finally, the last Section 8 concludes the research and resumes the work done.

## 2. Materials and Methods (Laboratory Tests)

This section details the methodology followed for carrying out the laboratory tests and provides the reader with some basic technical background on the Qorvo kit.

As already mentioned, two different testing campaigns were developed using Qorvo MDEK1001 kit. The first one was carried out in a laboratory environment, with the aim of assessing the pure performances of the RTLS system in controlled conditions. Specifically, the performance of the system was evaluated in terms of the following parameters: positioning accuracy (both in static and dynamic contexts); time to fix the position; sensitivity of tag/anchor antennas to orientation. The second campaign was instead designed and performed in an Italian company manufacturing mechanical plants in manual assembly lines, and aimed at assessing the RTLS requirements (in terms of number of anchor points, tags and their positioning) and its performance, computed as the ratio between successfully position determinations and total trials, for different tagged products and anchor configurations.

From a technical point of view, the Qorvo MDEK1001 kit (https://store.qorvo.com/datasheets/qorvo/mdek1001systemusermanual-1.1.pdf, accessed on 12 December 2022) is intended for letting the developers evaluate the RTLS modules and includes 12 multi-function DWM1001 boards (https://store.qorvo.com/datasheets/qorvo/dwm1001-devdatasheet-1.2.pdf, accessed on 12 December 2022). The DWM1001 module is based on the DW1000 UWB transceiver IC, which integrates UWB and Bluetooth antennas, RF circuitry, Nordic Semiconductor nRF52832 (https://www.nordicsemi.com/products/nrf52832, accessed on 12 December 2022) and a motion sensor. The antenna is vertically polarized, has an omnidirectional radiation pattern, a peak gain of 2.5 dBi and works in a frequency range of 5.5 to 7.5 GHz. DW1000 operates on UWB channel 5 (center frequency of 6.5 GHz), providing a data rate of up to 6.8 Mbps, and covering a range of 30 m. As a real-time solution, the kit is able to update the location of each tag in the network with a maximum frequency of 10 Hz.

Each module can be easily configured as an anchor (reference point whose coordinates are known), tag (moving item whose coordinates are computed by the system), listener (receiver logging RTLS data to a PC) or bridge (for the RTLS UWB–LAN gateway), by using the serial-over-USB link to a PC or using the ad hoc developed Android application (Figure 1), exploiting the integrated Bluetooth communication. While the anchors represent the fixed reference nodes in the system, a tag represents a mobile node.

The Qorvo RTLS system estimates the tags’ locations using trilateration. This mathematical technique uses the estimated range measurements between the target location (tag) and the closest three known locations (anchors) to determine the position of the tag relative to the known locations of the anchors [30]. Figure 2 illustrates the trilateration technique. Three or more anchors are located at known coordinates (*x_n_, y_n_*), while the tag is placed at coordinates (*x, y*), which are unknown and must be determined. The distance between the tag and the *n*-anchor is *d_n_*. For the system described, Equations (1)–(5) can be used to determine the coordinates of the tag (*x, y*) [31].
(1)$$x=\frac{\left(y_{2}-y_{1}\right)\gamma_{1}+\left(y_{2}-y_{3}\right)\gamma_{2}}{2\left[\left(x_{2}-x_{3}\right)\left(y_{2}-y_{1}\right)+\left(x_{1}-x_{2}\right)\left(y_{2}-y_{3}\right)\right]}$$
(2)$$y=\frac{\left(x_{2}-x_{1}\right)\gamma_{1}+\left(x_{2}-x_{3}\right)\gamma_{2}}{2\left[\left(x_{2}-x_{1}\right)\left(y_{2}-y_{3}\right)+\left(x_{2}-x_{3}\right)\left(y_{1}-y_{2}\right)\right]}$$
(3)$$\gamma_{1}=x_{2}^{2}-x_{3}^{2}+y_{2}^{2}-y_{3}^{2}+d_{3}^{2}-d_{2}^{2}$$
(4)$$\gamma_{2}=x_{1}^{2}-x_{2}^{2}+y_{1}^{2}-y_{2}^{2}+d_{2}^{2}-d_{1}^{2}$$
(5)$$d_{i}=\sqrt{{\left(x_{i}-x\right)}^{2}+{\left(y_{i}-y\right)}^{2}}\;i=1,\;2,\;3$$

When determining the position of the tag according to the above-mentioned equations in the UWB-WSN system, one of the most accurate mechanisms is the two-way-ranging (TWR) based on the IEEE 802.15.4a standard [16], which involves a two-way exchange of messages between the tag and each reference point (anchor). Typically, the tag begins the communication and sends the first message at time *T*_0_. The anchor replies at time *T_reply_* with a message containing the timestamps *T*_1_ and *T*_2_, which are the reception moments of the first packet and the reply to the tag, respectively, which are unknown to the mobile node (tag). At this point, the tag can compute the time of flight, which is in turn used to estimate the distance between the tag and the anchor, assuming the propagation speed to be known.

The basic setup requires four anchors to be placed approximately at the corners of the covered area (preferably rectangular), the moving tag(s) and a listener connected to a PC over a USB connection to configure the main parameters of the nodes and to log data. This topology is illustrated in Figure 3. The dashed lines between the anchors and the tag denote the UWB wireless connections.

### 2.1. Laboratory Preparation

For carrying out the first set of experiments, a suitable laboratory was identified in FabLab, a Fabrication Laboratory located inside the University of Parma. This facility, whose dimensions are approximately 11.3 × 5.7 m, is large enough to effectively complete the experiments and the shape factor is approximately the same as that of a common industrial building. The lab consists of a couple of rooms and a corridor between them.

A regular 2 × 2 m grid was traced on the floor all over the lab using paper tape as marker; 39 measuring points were marked, with 26 of them located in the main room. The position of the measurement points was determined using professional tools, such as a 7.5 m metal tape measure and a self-tuning crossline laser level. In the static tests, the tag was mounted on a tripod at a 1 m height, using a plumbline as a vertical reference line and exact positioning above the marked points of measurement.

Since the tests showed a higher read error in the second room (points 27 to 39), because of its irregular shape and to the presence of several walls along signal paths, this room was intentionally excluded from subsequent experimentations; hence, the testing campaign was limited to the main room (points 1 to 26).

Figure 4 below shows the plan and some pictures of the laboratory environment.

Once the grid was completed, each point was numbered, and its coordinates imported into a Microsoft Excel^TM^ spreadsheet. In addition, a further four reference points named A, B, C and D, placed approximately at the edges of the main room area, were chosen and marked with a red triangle for the installation of the anchors; their coordinates are thus known and can be used for determining the location of the tag under examination in the measurement points. The origin of the x- and y-axes was chosen at the bottom left corner of the laboratory, as shown in the map of Figure 4 above, close to the anchor point A. Anchors are always turned on and thus they are externally powered by a micro-USB socket to avoid the need for battery; because of the antenna polarization, the anchors were positioned vertically, facing each other in an elevated position (2.6 m), so as to avoid or limit the interference of objects placed along the line-of-sight between anchors and tag tested. During the execution of the test, it was decided to neglect the z-coordinate computed by the tags since it is not relevant in industrial environments.

During the measurements, the tag was positioned in the previously determined points; the tag’s antenna was kept parallel to anchor’s antenna in all the tests unless otherwise specified. The Microsoft Excel^TM^ program was considered the most suitable solution to collect, gather and post-process the data.

Table 1 below shows the coordinates of the four anchors, while Table 2 shows those of the reference points.

### 2.2. Experimental Campaign and KPIs Definition

In the following paragraphs, the KPIs chosen for the evaluation of the performance are detailed.

#### 2.2.1. Static Accuracy

Checking the static accuracy is important because in industrial use cases, the assets and the semi-finished products (also called Work In Progress, WIP) can be easily tracked inside a facility by using RTLS tags, enabling smart and more efficient picking processes. This indicator was used to assess the accuracy of the system in determining the tag’s position in stationary condition, to validate the capability of the tag to track industrial goods; tracked items can be effectively detected in the facility if the error is comparable to their size (typically 1 m for industrial assets and trolleys).

To this extent, the error between the position detected by the RTLS and the known true position was calculated; data were collected by placing the RTLS tag in all the measurement points of the main room according to the previously described methodology acquiring 100 position’s points at 10 Hz. Although the z-coordinate was ignored, the tag was kept at the same height of 1 m.

#### 2.2.2. Response Time (Dynamic Accuracy)

This indicator was used to evaluate the dynamic accuracy in determining the position of the tag in motion, to validate the capability of the tag to track industrial goods while they are handled in the facility by means of forklift trucks, conveyors or other material handling equipment. Tagged items can be effectively mapped in the facility if the delay in the response of the tag determines an error in the position of the item which is comparable to its size (again, typically 1 m for industrial assets and trolleys).

Dynamic accuracy was evaluated using a metal trolley as support for the tag; the choice was motivated by the fact that metal trolleys are very often adopted in industry for parts or as toolbox. An “L-shaped” trail was defined in the lab connecting points 1, 12, 15, 16, 17, 18 and 19; the moving tag was placed on a drawer in three different positions and on the top of the metal trolley as shown in Figure 5, in order to assess the position achieving the best performance. The operator moves the trolley along the defined route from point 1 to 19 and vice versa in about 38 s. The test included ten repetitions for each of the four positions of the tag on the trolley. The first position is in the center of the drawer, the second in close contact with the wall of the trolley, the third as far outwards as possible, and in the last position, the tag was placed on the top of the trolley. A maximum frequency of 10 Hz was defined as adequate for this experiment.

#### 2.2.3. Sensitivity to Relative Orientation

This third indicator was used to evaluate the influence of tag–anchor antenna orientation (either parallel or perpendicular) on the accuracy of the item’s determined position, to validate the capability of the tag to track industrial goods while they are positioned and handled in the facility at different orientation with respect to the anchors. Again, tagged items can be usefully detected in the facility if the error is comparable to their size, no matter their orientation.

The test procedure was the same as that of the previous test: the tag was placed on the top of the trolley and its antenna was set at a relative orientation of 0 degrees and 90 degrees with respect to the anchors’ antennas. The timestamp of the readings was neglected while the sampling frequency was set at 10 readings per second.

The test was repeated ten times for every different antenna’s orientation; the route was travelled in both directions, from point 1 to 19 and vice versa, for mitigating possible effects due to the presence of the operator.

#### 2.2.4. Start-Up Time

This last indicator was introduced to evaluate whether (and to what extent) the item position’s accuracy increases in time after a reset of the tag; this test does not have a direct industrial relevance for the specific use case but can be helpful to confirm and support the behavior of the tag assessed by the dynamic accuracy test in Section 2.2.2. The delay in the tag response in motion is expected to be comparable to the time required to fix the best position after a tag reset.

Four measurement points were used for this test, namely points 2 and 22 (wall proximity), and points 5 and 9 (center of the lab). The experiment was repeated ten times for each position; after a tag reset the position was logged for 30 s at the maximum frequency of 10 Hz, to point out the increasing accuracy in determining the position.

## 3. Results (Laboratory Test)

### 3.1. Static Accuracy

Table 3 below shows the results of the static accuracy test with respect to the main room; it is worthy pointing out that the overall average error is always below 1 m for each point and thus compatible with the specific use case investigated, i.e., location and inventory of industrial assets in a manufacturing facility. In the center of the laboratory, the error is minimum, while, as expected, it increases along the border walls because of the interferences caused by the building.

For completeness, Figure 6 graphically shows the plot of the positioning error for each acquired sample, while Figure 7 reports the average outcomes; Figure 8 then shows the average positioning error map.

Overall, the results are positive and promising for the specific industrial case study; indeed, the error is acceptable when using the technology for identifying industrial assets (trolleys, shelves, drawers and pallets) commonly used in manual assembly stations or for storage, as the size of such assets are widely greater than the error given by the RTLS system. It is possible to locate WIP and other products within the company with a high degree of accuracy, with the possibility of knowing their location without having to search for them.

### 3.2. Response Time (Dynamic Accuracy)

The second experiment has a very practical implication, since the tag was mounted on a metal trolley and moved in the main lab area.

The obtained results show that the position accuracy is very good along the entire route, according to the results of the previous test, as well as the delay in computing the position counting about 1 s.

Table 4 reports the average delay for each point along the route and for each position of the tag on the trolley (above or in the drawer) while Figure 9 plots the same data. As it can be noticed, the variance in the delay according to the placement of the tag is quite limited; in fact, the maximum variance between the best and the worst scenario is within 20%. In any case, the delay is compatible with the specific industrial use case considered, i.e., the identification of expensive spare parts, industrial assets, WIPs and assembly trolley or toolbox; on the other hand, the delay is not suitable for using this RTLS system to guide autonomous vehicles (LGVs or AGVs).

For completeness, Figure 10 below plots the real route taken in the lab (yellow line) during the execution of the experiment and the routes computed by the RTLS system in the best (light blue line) and the worst performing case (blue line). The blue and light blue bullet points represent the coordinates estimated by the RTLS tag, while the corresponding trolley location is given by the closest yellow bullet point; the distance between the yellow and (light) blue points is proportional to the delay given by the RTLS system.

### 3.3. Sensitivity to Orientation

As already stated, this test was aimed at assessing the potential influence of the combination tag–anchor antennas orientation on the RTLS performance, with a particular focus on the position accuracy. According to the lessons learned from the previous test, only one position of the tag on the trolley has been tested, namely the above position, which showed the minimum delay and thus, very likely, allows the best tag–anchor UWB communication.

Figure 11 plots on a dispersion chart the cloud of points measured by the RTLS system during the execution of the tests, providing a qualitative overview of the results; more in detail, the best and worst cases for each tag position (parallel and perpendicular) have been represented. Quantitative data are reported in Table 5; in that table, the average value and the variance of the error are computed for the parallel and perpendicular positions of the tag’s antenna. Compared to the perpendicular position, the parallel position shows a lower error, couped, however, with a higher variance; nonetheless, an overall view of the absolute values points out that the influence of the antenna’s relative position is negligible since the average error is, in both cases, comparable to the average accuracy error computed in the first experiment.

### 3.4. Start-Up Time

This test was aimed at assessing the increasing accuracy in the determination of the tag’s position as time elapses after a reset of the tag; in particular, the test point returns the time required by the RTLS tag to estimate the final coordinates of its position. These coordinates do not necessarily represent the real tag location, because of the reading error, as commented in the previous tests.

Figure 12 and Figure 13 plot the trend of the error made when computing the tag’s position during the first 2.5 s, using a linear and a logarithmic chart for each measurement point considered; the error is defined as the difference between the current computed coordinates of the tag and the final coordinates determined after 30 s of readings.

Table 6, instead, reports analytic data for each point; the values in bold refer to the time required by each point to reach an error lower than 10 cm.

## 4. Materials and Methods (In Situ Tests)

In this section, the testing activities performed in a real warehouse are described.

### 4.1. Facility Preparation

Test and measurements were carried out in the production plant of one of the major companies in Parma (North of Italy), leader in the production of machines for filling, packaging, palletizing/depalletizing in industrial sectors such as food, beverage, home and personal care and other industrial automation fields. The company’s plant provides an environment dedicated to the assembly of machines and production lines (assembly department), but the single components are stored in special warehouses, in predetermined locations. Employees equipped with forklifts trucks are in charge of picking up the various components and taking them to the assembly area, according to a warehouse management and control system. However, due to multiple situations, operators are not always able to perfectly follow the instructions assigned to them; as a result, it may happen that the components are not in the position they are expected to be, and the task of searching for the out-of-position components can cause a significant waste of time and financial resources.

To avoid delays and economic damage due to the loss of components, the company is interested in tracing the position of the most strategic and expensive components, and this is the reason why a partnership with the University of Parma was established. The map of the warehouse is illustrated in Figure 14; inside the warehouse, coverage of partial areas used for various tests has been identified.

A proper set up was realized to carry out the test; in particular, anchor devices were placed on 3 m height poles as shown in Figure 15.

In Figure 16, a photo of a view of the warehouse in which the test activities were carried out is proposed.

The listener device was connected to a PC through a USB cable emulating a serial connection, to collect the information of the relative coordinates of the tag as computed by the RTLS system. Collected data were recorded via a text file, then arranged and analyzed with Excel^TM^.

### 4.2. Experimental Campaign and KPIs Definition

Test were implemented according to six different configurations presented below in Table 7. The purpose of the tests is to verify the dependence of the performance of the RTLS system with respect to some parameters: the covered area, the number of anchors and the presence of obstacles such as concrete pillars and metal shelves. The performance of the system is computed as the rate between the successful position acquisition and the total acquisition trials.

Tests 1–5 are aimed at evaluating the best arrangement and number of anchors in the industrial building; related details are proposed in Table 7. During the execution of tests, the tag is moved by an operator who holds it in his hand at the height of about 2 m; again the z-coordinate is neglected, being out-of-scope.

Test 6, instead, is intended for comparatively evaluating the different performance levels of the position of the tag on a metal asset and its details are presented in Table 8 below. This test was designed to evaluate the effects of the tag position on the performance of the RTLS system in movement; three tags were installed on a mobile metal shelf and placed in three differing positions (high, medium and low height).

In the subsections that follow, each testing session is briefly proposed.

## 5. Results (In Situ Tests)

### 5.1. Test 1

The first test involved a partial coverage of the warehouse area, represented in Figure 15. The area of interest is roughly 21 m × 28 m (approximately 600 m^2^) and is characterized by the presence of five concrete pillars, represented by black-square within the area of interest. Five RTLS devices were used: four fixed anchors, represented by red triangles in Figure 17, and one mobile tag.

Figure 18 shows the real-time movements of the single tag acquired by the RTLS systems. The dotted line shows the real route of the tag, while the solid points represent the coordinates of the tags detected by the RTLS.

### 5.2. Test 2

The second test was very similar to the first (four anchors and one tag) and was carried out in the same area of interest (see Figure 17), characterized by the presence of five concrete pillars. The difference with the first test concerns the mobile tag route reported in Figure 19.

### 5.3. Test 3

In the third test, the area of interest was reduced to 21 × 14 square meters; with this layout, the measurement of the position by the RTLS was not affected by the presence of pillars, since they were positioned on the perimeter of the area of interest. The results of this test are shown below in Figure 20, in which the path followed by the tag is drawn with a line and the gathered points are marked in blue. Despite the absence of pillars, even in this case, the measurements acquired by the RTLS system do not completely fit the path followed by the tag.

### 5.4. Test 4

Test 4 was carried out on a large area of 41 × 13 square meters (Figure 21), without pillars. As in the previous test, the RTLS consisted of four anchors and a single tag. Again, there were trajectories covered by the tag for which it was not possible to have correct data acquisitions by the RTLS system as shown in Figure 19.

### 5.5. Test 5

Test 5 was carried out on an area of 41 × 13 square meters in the same area selected for Test 4 but, this time, the RTLS system was composed of six anchors instead of four, as shown in Figure 22, which plots also the trajectory made by the tag during its movements. As expected, the presence of six anchors clearly improves the ability of the RTLS system to identify the correct position; in fact, the dotted line is almost completely covered by the solid blue points.

### 5.6. Test 6

Test 6 has the same characteristics as the previous test, although three tags are simultaneously used and installed on a mobile metal shelf in three different positions. These configurations are called High-Tag, Mid-Tag and Low-Tag, respectively, as shown in Figure 23. This test was designed for evaluating the effects of the tag height on the performance of the RTLS system.

The three figures below (i.e., Figure 24, Figure 25 and Figure 26) show the trajectories of the tags detected in Test 6 as a function of their position: High-Tag (green line), Mid-Tag (red line) and Low-Tag (blue line).

## 6. Discussion

According to the results of the presented tests summarized in the following Table 9, some lesson learned can be derived:The performance of the RTLS is very sensitive to the presence of concrete pillars;Increasing the number of anchors within the same covered area improves the success rate of the localization system.

Tests 1–4 point out that when using four anchors, there are several areas where the system is unable to locate the tag; in the worst case there is a (low) 40% success rate. Using six anchors instead of four, the success rate increases up to 90%. The accuracy of the position is not measured as already assessed in laboratory tests.

As regards to the sensitivity of determining the position of the tag with respect to its positioning on the movable metal shelf, it can be observed that determining the position of a High-Tag has a much higher success rate than in the remaining two cases (Mid- and Low-Tag), as inferable from Table 10 which proposes the results from Test 6. The lower success rates of the medium and lower tag show that the RTLS system significantly degrades its performance in the presence of objects that shield the electromagnetic signal between the tag and the anchors.

Overall, it can be stated that all the experimental observations are in line with what could be expected.

## 7. Economic Assessment

Given the purpose of the research, namely that of proposing a low-cost solution, for demonstrating the economic suitability of the system, an economic analysis was made. Specifically, a market survey was performed; characteristics and related costs were determined and compared for two additional brands among the most renown and reliable on the market according to the opinion of the authors, namely Zebra and Siemens.

The resulting comparison is proposed in the following Table 11. Note that, according to the results of the testing phase, it was reasonable to assume a number of anchors equal to six (properly multiplied by two for a complete coverage of the warehouse) and a number of tags equal to 100 for the WIP and assets.

As it is possible to deduce from the last two rows, the cost for installing RTLS like the one designed and tested in this manuscript is significantly lower than the other two solutions. More precisely, the cost for the infrastructure is fifty times less than the Zebra solution, and almost forty if compared with Siemens. Great savings as well occur when considering the cost for the tags.

The absolute cost-effectiveness of this solution if therefore fully justified.

## 8. Conclusions

This paper has dealt with the development and subsequent testing of a low-cost RTLS solution based on UWB signals, with the aim of assessing its potential application in a real industrial environment. The proposed system relies on the Qorvo MDEK1001, a system usually implemented for indoor positioning available on the market. The role that this system is supposed to have in an industrial context is that of instantly tracking assets within indoor environments, which is often difficult and challenging, given the presence of walls, pillars, metal shelves and moving objects of different materials. To this extent, specific industrial products are proposed by the major producers of Auto ID technologies, but at very high price, which involves high adoption barriers and prevents the possibility of small companies (SMEs) to implement RTLS tracking.

The solution was firstly tested in a laboratory at the University of Parma, and this preliminary phase allowed for the assessment of its static accuracy, response time, sensitivity to orientation and start-up time. In summary, the average position accuracy was lower than 1 m in any test and the delay in the position response was lower than 2 s; this performance makes the system a suitable candidate also for industrial asset tracking.

The subsequent testing session took place in a real industrial context, namely the warehouse of an Italian manufacturing company operating in Parma (Italy). Five tests were performed with different characteristics in terms of field sizes and number of anchors, for determining the best configuration (number of anchors and their positioning). A sixth additional test, finally, aimed at evaluating the effects of the tags’ position on the item, with reference to the quality of measurement by the RTLS system. Once the coverage of the area of interest is well performed by an adequate number of anchors (approximately arranged in a 20 × 20 m grid), the reading success rate varies within a range between 61% and 97% according to tag placement on the metal asset, confirming the potential adoption of this technology.

The economic assessment returns very interesting figures as well; specifically, a typical configuration of the system, covering a 41 × 28 m industrial building and using 100 tags for assets tracking, generates an investment which is one tenth of the cost of similar solutions developed by competitors and available on the market.

From a practical perspective, some further steps will be required for a full deployment of the Qorvo solution in an industrial environment, given the great performance of the system perfectly matching asset tracking needs. Firstly, the case of the module has to be reengineered and ruggedized (and then tested again) to resist to mechanical shocks and to facilitate the application of the tag to the metal asset. Moreover, a testing session in a multi-tag configuration will be required for confirming the capability of the system to manage as many as about 200 tags in the same covered area [29]. Lastly, the tag’s battery lifetime deserves a dedicated testing campaign, to assess its compatibility with the industrial use case; the Qorvo estimation tool reports an expected battery duration of 500 days working at a 0.1 Hz report rate and using a rechargeable 3.7 V li-ion RCR123 battery (600 mAh). The reengineering of the tag enclosure may also encompass the possibility to install a bigger li-ion battery with higher capacitance.

Overall, this paper may be useful for practitioners who intend to adopt similar solutions, as it suggests the steps to be followed, performance indicators to be assessed and measured and the correct procedure for a real implementation; the in-field study is also the element that characterizes this manuscript and makes it different from other research reports in the field.

Moreover, the real attractiveness of this system is the extreme affordability of the investment, as demonstrated by the results of the economic comparison. This may be a cue for other researchers.

## Figures and Tables

**Figure 1 sensors-23-01124-f001:**
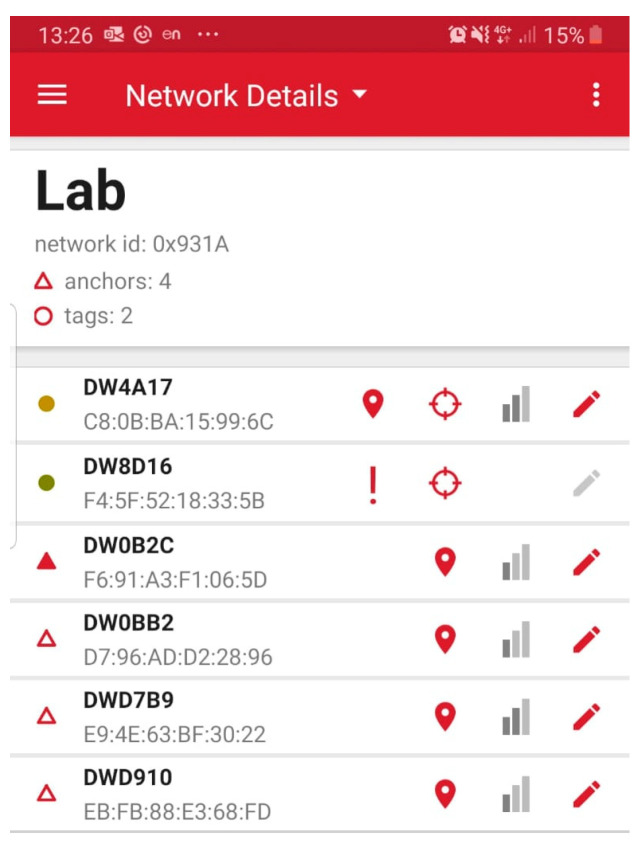
Lab RTLS network configuration using Android application.

**Figure 2 sensors-23-01124-f002:**
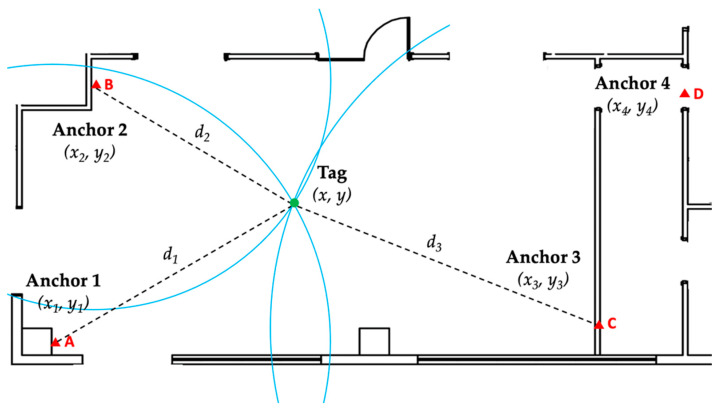
Illustration of trilateration technique.

**Figure 3 sensors-23-01124-f003:**
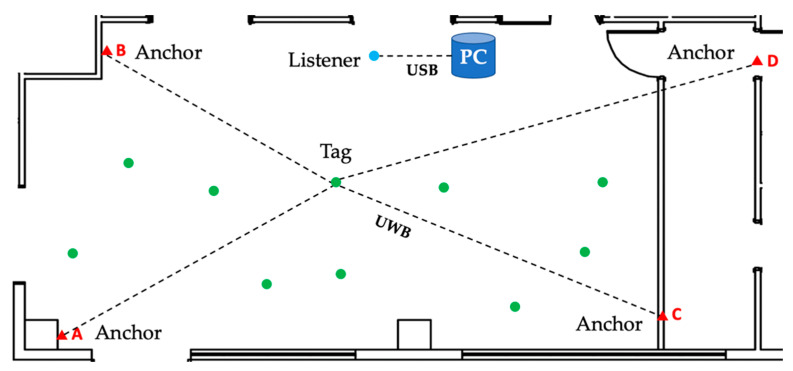
Topology of the RTLS network.

**Figure 4 sensors-23-01124-f004:**
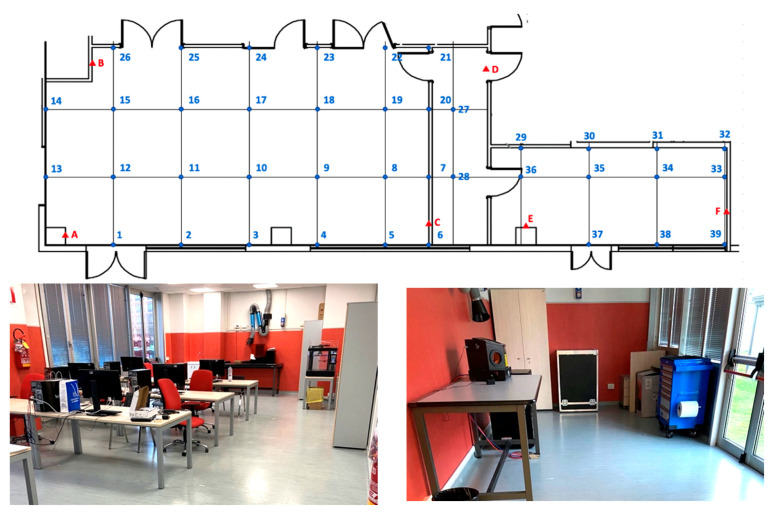
Map of the area (**above**) and two pictures of the laboratory, the main room (**bottom left**) and the secondary room (**bottom right**).

**Figure 5 sensors-23-01124-f005:**
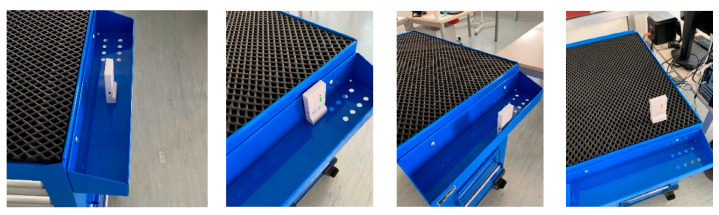
Positions of the moving tags (**middle**, **internal**, **external**, **above**).

**Figure 6 sensors-23-01124-f006:**
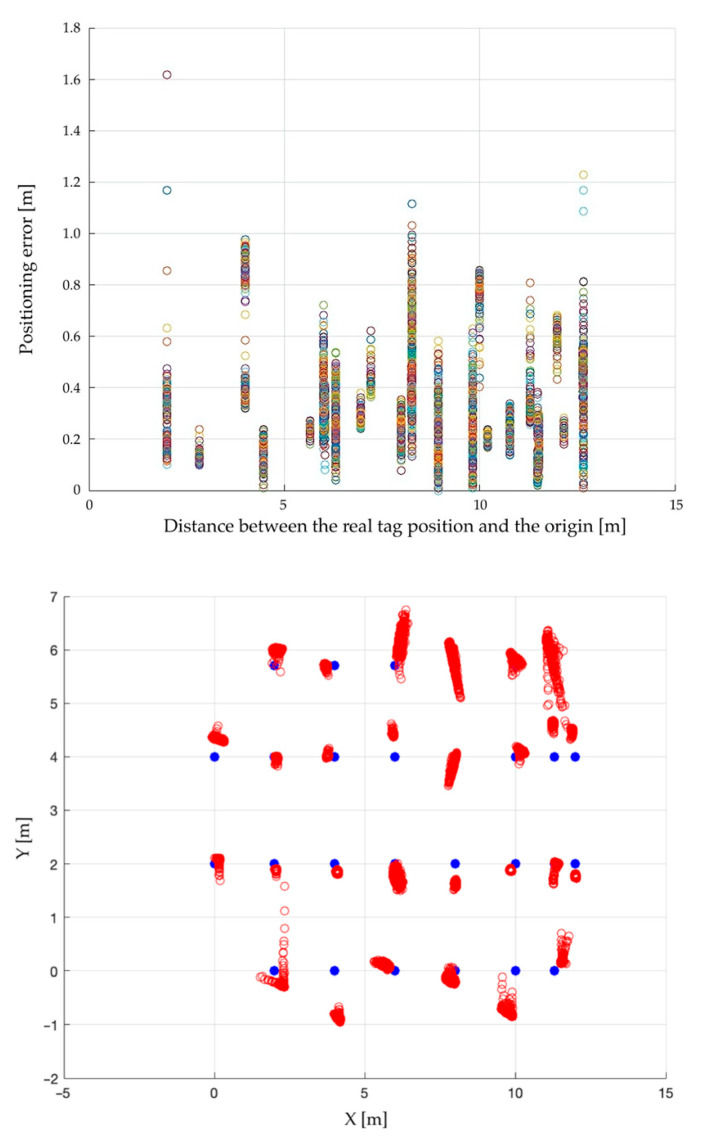
Positioning error plot for each sample (**above**) and map (**below**).

**Figure 7 sensors-23-01124-f007:**
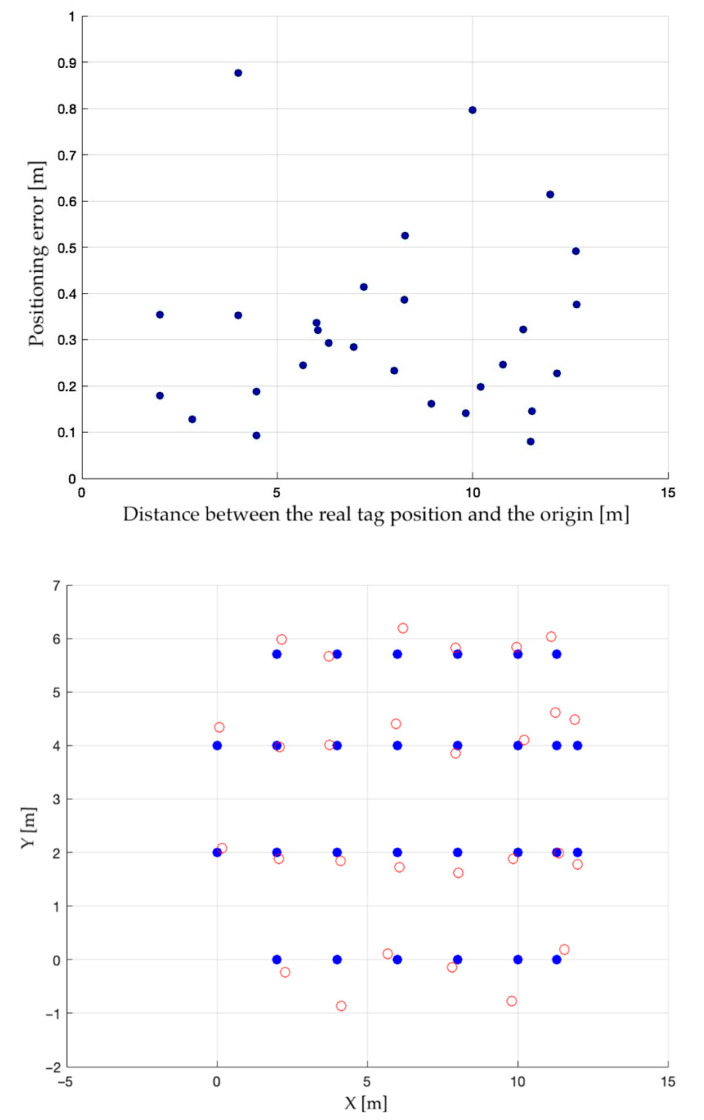
Average positioning error plot (**above**) and map (**below**).

**Figure 8 sensors-23-01124-f008:**
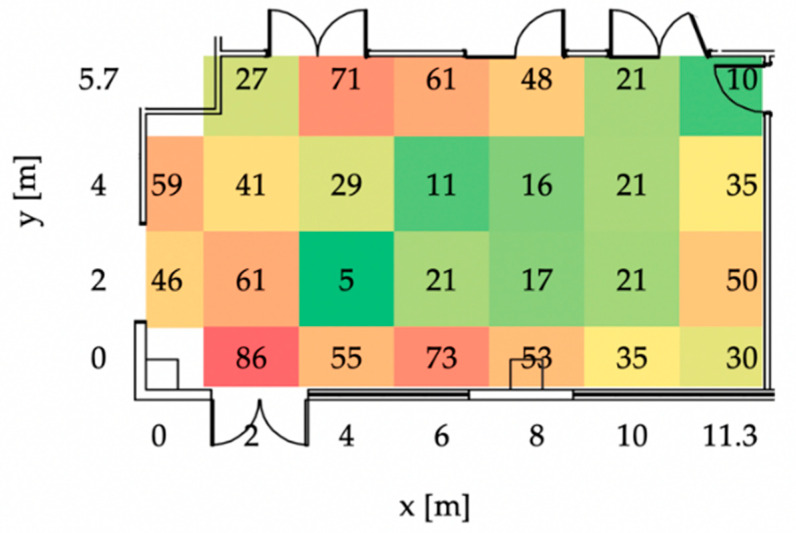
Average positioning error map.

**Figure 9 sensors-23-01124-f009:**
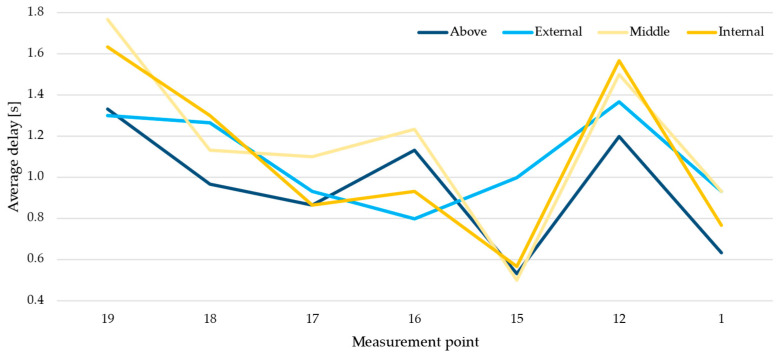
Recorded average delays, according to the tag position.

**Figure 10 sensors-23-01124-f010:**
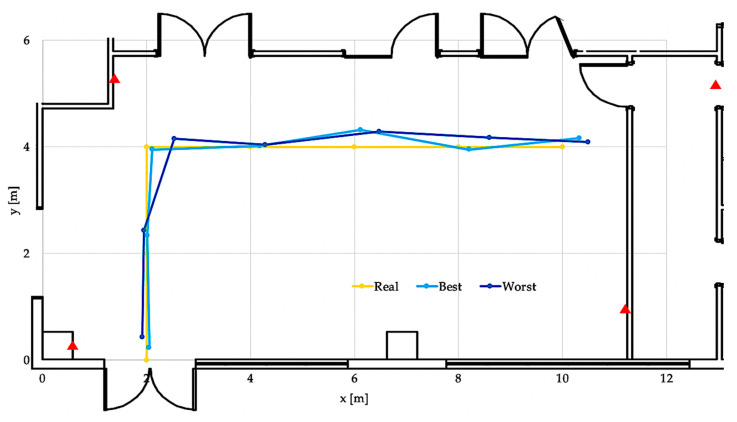
Real, best and worst routes computed by the RTLS system.

**Figure 11 sensors-23-01124-f011:**
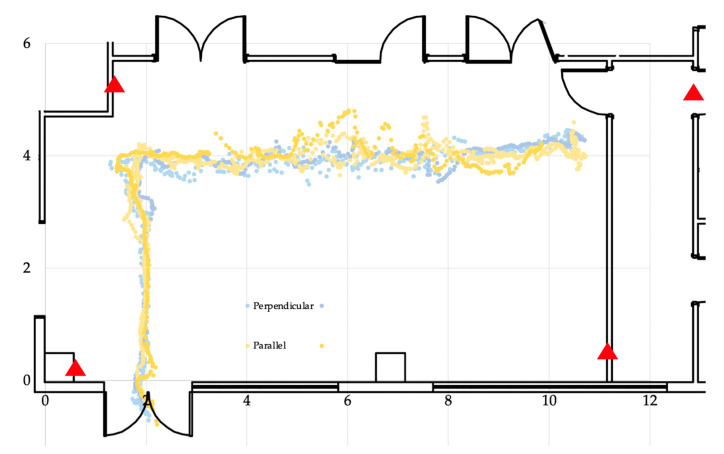
Best and worst cases for each tag position.

**Figure 12 sensors-23-01124-f012:**
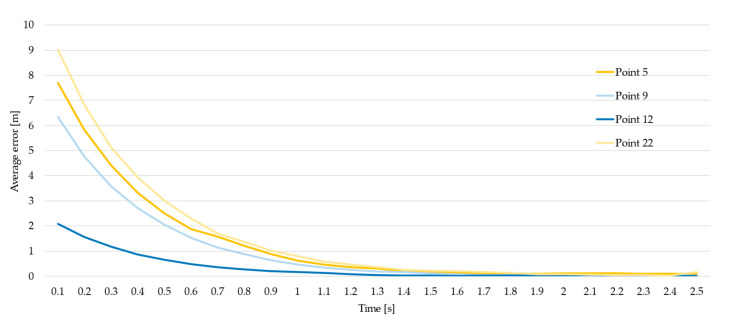
Error in the first 2.5 s (linear).

**Figure 13 sensors-23-01124-f013:**
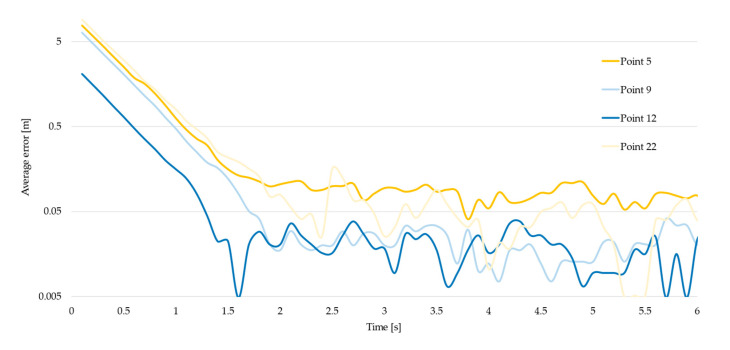
Error in the first 6 s (logarithmic chart).

**Figure 14 sensors-23-01124-f014:**
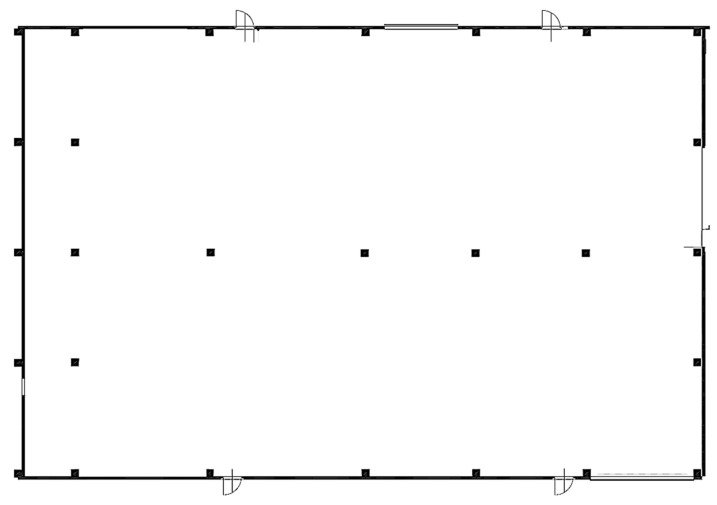
The plan of the warehouse.

**Figure 15 sensors-23-01124-f015:**
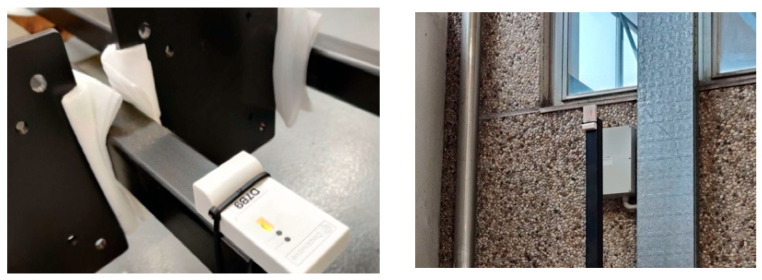
Anchor devices fixed on the edge of the pole.

**Figure 16 sensors-23-01124-f016:**
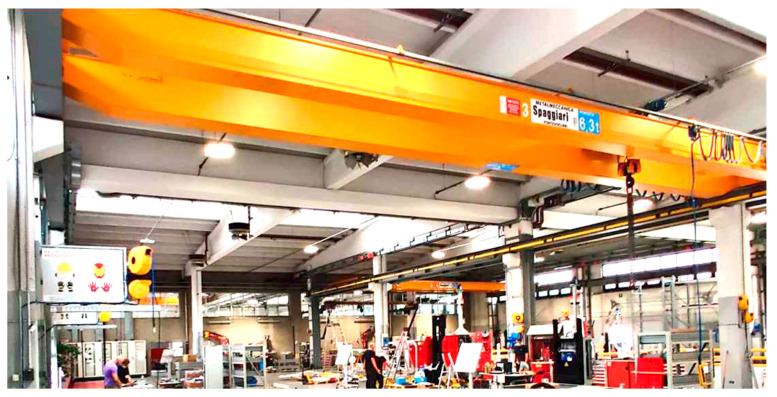
A view of the warehouse used for testing activities.

**Figure 17 sensors-23-01124-f017:**
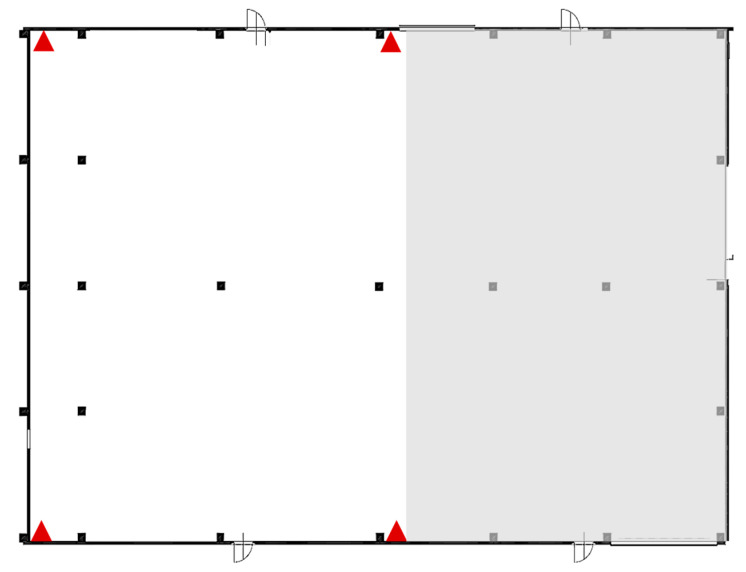
Area of interest used for Test 1.

**Figure 18 sensors-23-01124-f018:**
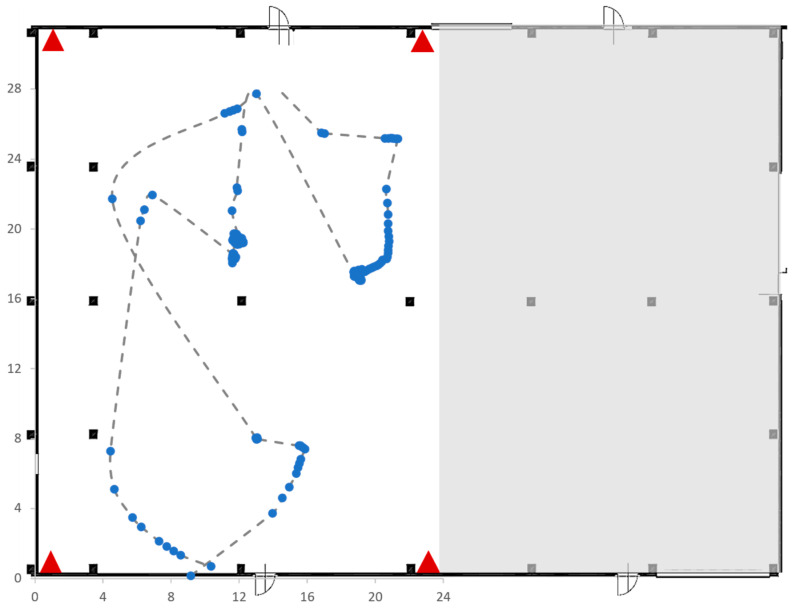
Trajectory followed by the tag in Test 1.

**Figure 19 sensors-23-01124-f019:**
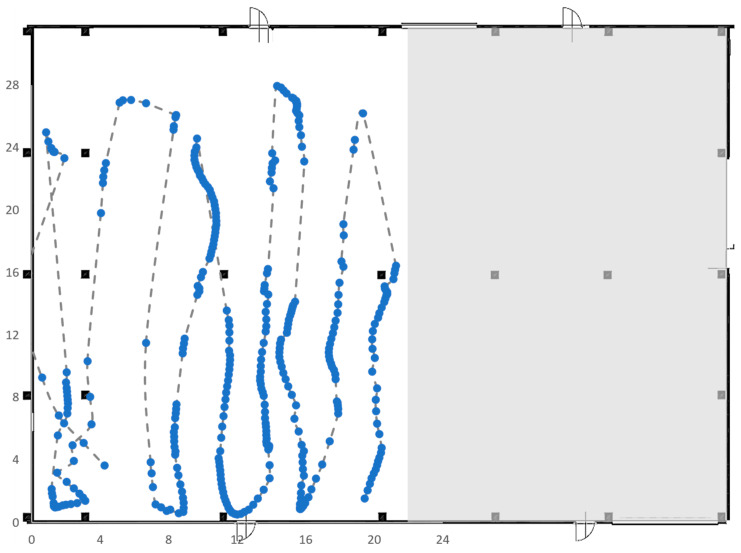
Trajectory followed by the tag in Test 2.

**Figure 20 sensors-23-01124-f020:**
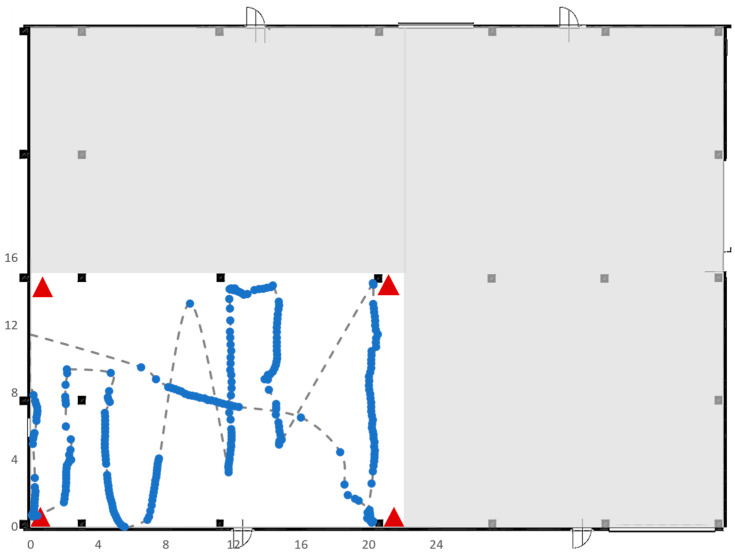
Trajectory followed by the tag in Test 3.

**Figure 21 sensors-23-01124-f021:**
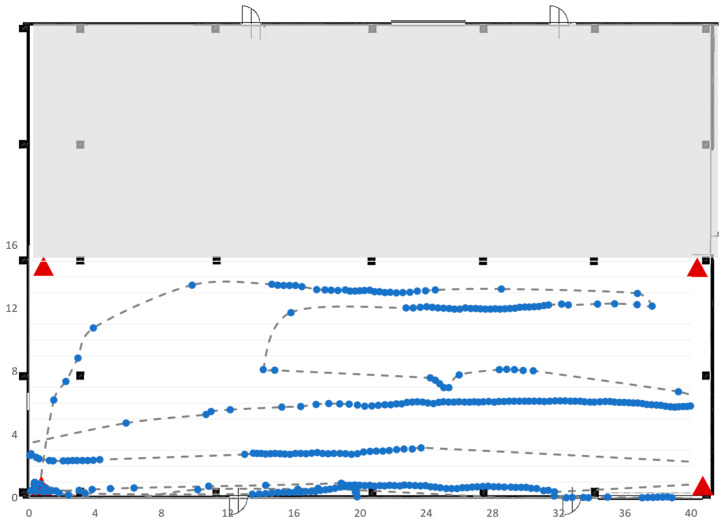
Trajectory followed by the tag in Test 4.

**Figure 22 sensors-23-01124-f022:**
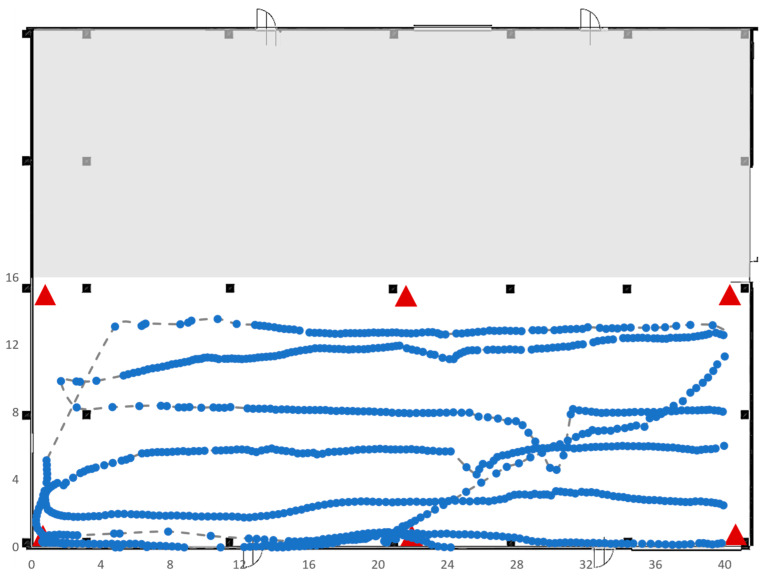
Trajectory followed by the tag of Test 5.

**Figure 23 sensors-23-01124-f023:**
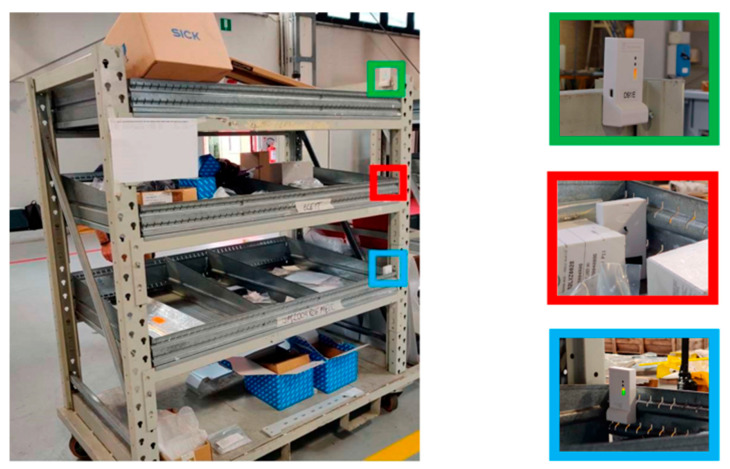
Different positions of the tag in Test 6: High-Tag (green), Mid-Tag (red) and Low-Tag (blue).

**Figure 24 sensors-23-01124-f024:**
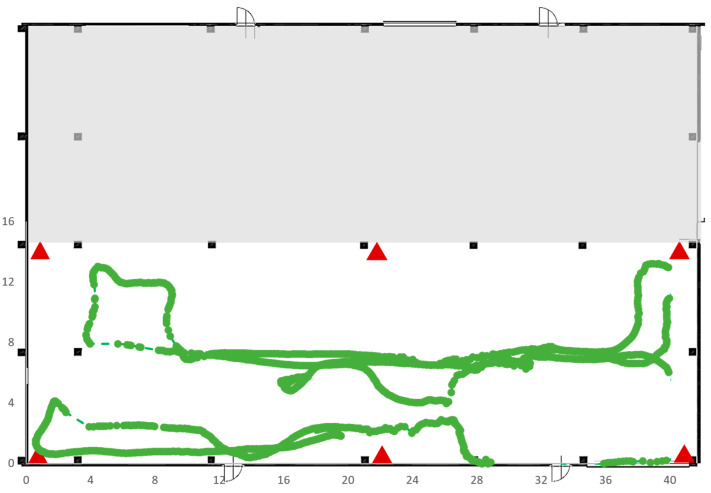
Results of Test 6; tag in the higher position.

**Figure 25 sensors-23-01124-f025:**
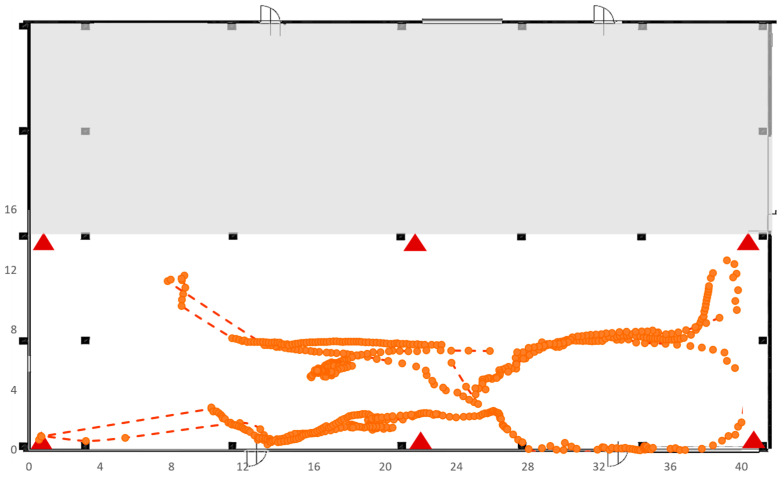
Results of Test 6; tag in the medium position.

**Figure 26 sensors-23-01124-f026:**
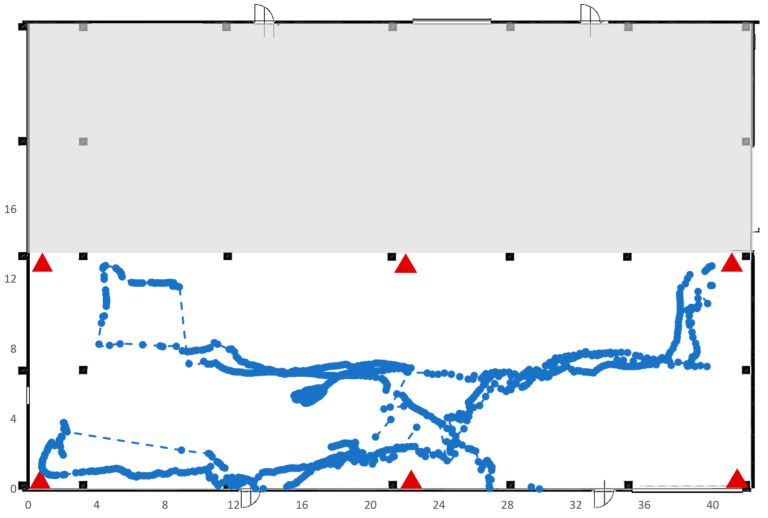
Results of Test 6; tag in the lower position.

**Table 1 sensors-23-01124-t001:** Anchors coordinates.

Anchor	ID	x [cm]	y [cm]	z [cm]
A	0BB2	47	25	264
B	0B2C	135	525	264
C	D79B	1130	60	264
D	D910	1300	502	264

**Table 2 sensors-23-01124-t002:** Coordinates of the reference points (tag positions).

Ref Point	x [cm]	y [cm]	Ref Point	x [cm]	y [cm]
1	200	0	21	1130	570
2	400	0	22	1000	570
3	600	0	23	800	570
4	800	0	24	600	570
5	1000	0	25	400	570
6	1130	0	26	200	570
7	1130	200	27	1200	400
8	1000	200	28	1200	200
9	800	200	29	1400	280
10	600	200	30	1600	280
11	400	200	31	1800	280
12	200	200	32	2005	280
13	0	200	33	2005	200
14	0	400	34	1800	200
15	200	400	35	1600	200
16	400	400	36	1400	200
17	600	400	37	1600	0
18	800	400	38	1800	0
19	1000	400	39	2005	0
20	1130	400			

**Table 3 sensors-23-01124-t003:** Static accuracy test summary results.

	x	y
Average error [m]	0.16	0.31
Percent with error less than 10 cm	43%	18%

**Table 4 sensors-23-01124-t004:** Average delay for each point as a function of the tag position.

Tag Position	Measurement Point							Avg. Delay [s]
	19	18	17	16	15	12	1	
Above	1.3	1.0	0.9	1.1	0.5	1.2	0.6	1.0
External	1.3	1.3	0.9	0.8	1.0	1.4	0.9	1.1
Middle	1.8	1.1	1.1	1.2	0.5	1.5	0.9	1.2
Internal	1.6	1.3	0.9	0.9	0.6	1.6	0.8	1.1
Avg. Delay [s]	1.5	1.2	0.9	1.0	0.7	1.4	0.8	

**Table 5 sensors-23-01124-t005:** Results from the sensitivity to orientation test.

Antenna Orientation	Parallel	Perpendicular
Error variance [m^2^]	0.070	0.067
Average error [m]	−0.023	0.037
Avg. x-axis error [m]	0.013	0.070
Avg. y-axis error [m]	−0.046	0.022

**Table 6 sensors-23-01124-t006:** Average error for each point.

Time	Average Error [m]			
	Point 5	Point 9	Point 12	Point 22
0.1	7.69	6.34	2.07	9.02
0.2	5.83	4.77	1.56	6.83
0.3	4.41	3.59	1.17	5.13
0.4	3.31	2.71	0.86	3.94
0.5	2.50	2.05	0.65	3.01
0.6	1.87	1.53	0.48	2.29
0.7	1.59	1.15	0.36	1.70
0.8	1.21	0.87	0.27	1.36
0.9	0.88	0.63	0.20	1.02
1.0	0.62	0.47	0.16	0.80
1.1	0.46	0.33	0.12	0.58
1.2	0.36	0.25	**0.08**	0.46
1.3	0.30	0.19	0.04	0.37
1.4	0.20	0.16	0.02	0.25
1.5	0.16	0.12	0.02	0.22
1.6	0.13	**0.08**	0.00	0.19
1.7	0.12	0.05	0.02	0.16
1.8	0.11	0.04	0.03	0.13
1.9	**0.10**	0.02	0.02	**0.08**
2.0	0.10	0.02	0.02	0.08

**Table 7 sensors-23-01124-t007:** Test 1–5 configurations.

Test n.	Field Size [m × m]	Anchors [n.]
1	21 × 28	4
2	21 × 28	4
3	21 × 14	4
4	41 × 13	4
5	41 × 13	6

**Table 8 sensors-23-01124-t008:** Test 6 configuration.

Test n.	Field Size [m × m]	Anchors [n.]
6	41 × 13	6

**Table 9 sensors-23-01124-t009:** Summary of results from tests 1–5 in terms of success reading rate.

Test n.	Field Size [m × m]	Anchors [n.]	Pillars	Success Readings	Total Readings	Success Rate [%]
1	21 × 28	4	Yes	620	1225	50.6
2	21 × 28	4	Yes	364	908	40.1
3	21 × 14	4	No	592	707	83.7
4	41 × 13	4	No	727	1135	64.1
5	41 × 13	6	No	1069	1149	93.0

**Table 10 sensors-23-01124-t010:** Summary of results from Test 6 in terms of success reading rate.

Tag Position	Success Readings	Total Readings	Success Rate [%]
High	1569	1614	97.2
Medium	761	1231	61.8
High	1270	1453	87.4

**Table 11 sensors-23-01124-t011:** Economic comparison, considering the whole warehouse area.

Brand &	Qorvo	Zebra	Siemens
Solution Name	MDEK1001	Dart UWB	SIMATIC RTLS
ID Target	Assets	People and Assets	People and Assets
Position Accuracy	10 cm	30 cm	20 cm
Tag/Reader distance	30 m	200 m	50 m
Different components	1 (Configurable as anchor, tag, listener, bridge)	4 (Hub, sensors, wand, tags)	2 (Gateways and transponders)
	Case study: 41 × 28 m^2^		
Reading spots name and number	Anchors: 12 units	Sensor: 4 units	Gateway: 4 units
Other infrastructure’s components	-	Hub: 1 unit Wand: 1 unit	-
Infrastructure costs	About 360 €	About 18,000 €	About 14,000 €
Asset tags (100 pcs.)	About 2000 €	About 10,000 €	About 20,000 €

## Data Availability

The data presented in this study are available on request from the corresponding author. The data are not publicly available due to privacy restrictions.

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
