# Peer review of "Low-Cost UWB Based Real-Time Locating System: Development, Lab Test, Industrial Implementation and Economic Assessment"

_sensors, 2023, doi:10.3390/s23031124_

Round 1

Reviewer 1 Report

The authors of this manuscript suggest a low-cost technical solution for a real-time location system (RTLS) for assets being moved around a production facility. The chosen technology is based on ultra-wideband (UWB) redio frequency (RF). The introduction section provides enough context and related literature that primes the reader to comprehend the situation and the state of the art. The second section describes in detail the materials and methods used in a lab environment, as well as the adopted topology of the RTLS network. This is aided by the addition of related figures and tables. The third section provides and discusses the computed numerical results, once again in a series of lab tests. The fourth section provides the activities performed in a real warehouse attached to a production plant. Detailed description of the area is provided here, along with images of the production plant. Once again, a series of tests are conducted and their results are adequately presented and discussed. A discussion of the proposed system is provided in section 5. Appropriate commentary is provided on the obtained results. Section 6 is the most interesting part of this manuscript, as it provides a comparative assessment among 3 different RTLSs. Those include the use of the proposed Qorvo unit, a Zebra model and a Siemens model. Table 11 showcases the superiority of the proposed system in terms of low costs. Finally, the conclusions of the manuscript are drawn in section 7, where the authors summarize their proposition, their tests and results, and the possibility of offering a viable solution to track assets within a production plant, at a low cost.

Overall, this manuscript offers a viable, practical, low cost solution. The flow of the manuscript is good and it is readable. However, some grammar, spelling, punctuation and unnatural use of language is detected. Thus, a thorough English language revision is highly encouraged before publication. A few examples of such linguistic mistakes include:

1. In line 145, the authors use the word “trilateration”. Did they actually mean “triangulation”? Also in the same line, they use the word “rage”. Did they actually mean “range”?

2. In line 240, grammar mistake, “…the tag was placed the on top of the trolley…”

3. In line 279, unnatural use of language in “… are absolutely brilliant for the …”

4. In line 321, the authors use the acronym “RTS”. Did they mean to use “RTLS” instead?

5. In line 375, the caption of Fig. 12 uses the word “plant”. Did the authors actually mean “plan”?

6. In line 376, the authors use the word “suited”. This word does not make much sense in the context of its use here.

7. In line 393, the section heading has a spelling mistake.

Mostly, the provided references are recent and somehow adequate. However, references [13] and [20] are incomplete and need to be corrected.

Author Response

Dear reviewers,

First of all, we would like to thank you for your time and care in reviewing our manuscript, as well as for all your valuable comments. We have tried to implement all them for improving the quality and the readability of the paper.

The structure of the paper is the same of the original submission, as all of you approved it. The main changes have been made in the introduction section and in the methodology (i.e., Material and Methods section): as requested, more contents were included, as well as additional technical information on the proposed system design and the architecture. Moreover, the paper was subjected to an accurate revision of the English language, as recommended.

Below you can find punctual detailed replies for each comment (in red), while in the main text changes related to the contents are yellow highlighted so as to be easily identified.

We hope that you will find the new version more suitable for publication.

Sincerely,

The authors

REVIEWER #1

The authors of this manuscript suggest a low-cost technical solution for a real-time location system (RTLS) for assets being moved around a production facility. The chosen technology is based on ultra-wideband (UWB) radio frequency (RF). The introduction section provides enough context and related literature that primes the reader to comprehend the situation and the state of the art. The second section describes in detail the materials and methods used in a lab environment, as well as the adopted topology of the RTLS network. This is aided by the addition of related figures and tables. The third section provides and discusses the computed numerical results, once again in a series of lab tests. The fourth section provides the activities performed in a real warehouse attached to a production plant. Detailed description of the area is provided here, along with images of the production plant. Once again, a series of tests are conducted and their results are adequately presented and discussed. A discussion of the proposed system is provided in section 5. Appropriate commentary is provided on the obtained results. Section 6 is the most interesting part of this manuscript, as it provides a comparative assessment among 3 different RTLSs. Those include the use of the proposed Qorvo unit, a Zebra model and a Siemens model. Table 11 showcases the superiority of the proposed system in terms of low costs. Finally, the conclusions of the manuscript are drawn in section 7, where the authors summarize their proposition, their tests and results, and the possibility of offering a viable solution to track assets within a production plant, at a low cost.

Overall, this manuscript offers a viable, practical, low-cost solution. The flow of the manuscript is good and it is readable. However, some grammar, spelling, punctuation and unnatural use of language is detected. Thus, a thorough English language revision is highly encouraged before publication. A few examples of such linguistic mistakes include:

  1. In line 145, the authors use the word “trilateration”. Did they actually mean “triangulation”? Also in the same line, they use the word “rage”. Did they actually mean “range”?

We confirm that is trilateration! With reference to this concept, we added more information, for completeness and for the sake of clarity.

  1. In line 240, grammar mistake, “…the tag was placed the on top of the trolley…”
  2. In line 279, unnatural use of language in “… are absolutely brilliant for the …”
  3. In line 321, the authors use the acronym “RTS”. Did they mean to use “RTLS” instead?
  4. In line 375, the caption of Fig. 12 uses the word “plant”. Did the authors actually mean “plan”?
  5. In line 376, the authors use the word “suited”. This word does not make much sense in the context of its use here.
  6. In line 393, the section heading has a spelling mistake.

Mostly, the provided references are recent and somehow adequate. However, references [13] and [20] are incomplete and need to be corrected.

Dear reviewer #1, thank you for your positive evaluation of our manuscript; we are glad that your appreciated it. As requested, the paper has undergone a substantial revision of the language, so as to remove mistakes (such as those you listed and others, which were properly corrected) and improve the readability. References 13 and 20 were completed. Thank you for your notes.

Reviewer 2 Report

1.      Proposed System Design and Architecture is not clear. Absent of details parameters of Proposed System Design and Architecture

2.      More relevant and recent references should be included in this work

3.      Mismatch of grammar of different sentences

4.      Conclusion should be more specific way that means. Purpose of this work is not clear? Outcome of the work should be more clearly highlighted in both abstract and conclusion section.

Author Response

Dear reviewers,

First of all, we would like to thank you for your time and care in reviewing our manuscript, as well as for all your valuable comments. We have tried to implement all them for improving the quality and the readability of the paper.

The structure of the paper is the same of the original submission, as all of you approved it. The main changes have been made in the introduction section and in the methodology (i.e., Material and Methods section): as requested, more contents were included, as well as additional technical information on the proposed system design and the architecture. Moreover, the paper was subjected to an accurate revision of the English language, as recommended.

Below you can find punctual detailed replies for each comment (in red), while in the main text changes related to the contents are yellow highlighted so as to be easily identified.

We hope that you will find the new version more suitable for publication.

Sincerely,

The authors

REVIEWER #2

  1. Proposed System Design and Architecture is not clear. Absent of details parameters of Proposed System Design and Architecture.

The description of the RTLS system has been improved according to your comment; in particular, section 2 “Materials and Methods (laboratory tests)” has been enriched with more details and improved figures (i.e., the new figures 1, 2 and 3) about the system and its configuration. We hope to have met your expectations.

  1. More relevant and recent references should be included in this work.

Despite the other two reviewers stated that the references were recent and adequate, new ones were included to comply with your request (yellow highlighted in the list at the end of the manuscript; 12 specifically).

  1. Mismatch of grammar of different sentences.

As requested also by other reviewers, the paper was subjected to a deep revision of the English, so as to remove mistakes and mismatches. Thank you for having pointed that out.

  1. Conclusion should be more specific way that means. Purpose of this work is not clear? Outcome of the work should be more clearly highlighted in both abstract and conclusion section.

The introduction and the conclusions have been amended according to the request and the aim of the study was better declared; the improvements have been highlighted within the text in yellow. Future research activities were also recalled at the end of the manuscript. Thank you for your revision!

Reviewer 3 Report

The author introduced about Low-cost UWB-based Real Time Locating System. 

The novelty and the proposed work are satisfied with the following comments.

# In the abstract, highlight the novelty by adding a few more features of the proposed model.

# In the introduction, suggested including the

-key points of node localization of wireless networks with respect to the proposed system

-Add a few more concepts of industrial wireless networks with respect to the proposed model

- Include a few more concepts related to Industrial WSN with RTLS 

refer to the links if needed:

https://www.hindawi.com/journals/wcmc/2022/5290028/

https://www.sciencedirect.com/science/article/pii/S2214785321017582

https://link.springer.com/chapter/10.1007/978-981-19-1844-5_64

# In Section 2, from 2.1.1 to 2.1.4 =>add the key points and highlight the importance with respect to the proposed model in order to match the following terms Static accuracy, (Dynamic accuracy), Sensitivity, and Start-up time.

The results with a graphical analysis of various parameters are satisfying the proposed idea.  

#In section 3, Figure 4 - clarity is not satisfied. Change with better quality.

#In section 4, suggested including some additional key points and highlighting the results by comparing the existing work from test 1 to test 6

# In Conclusion, add the extension of future work by highlighting the limitations of the proposed work

Author Response

Dear reviewers,

First of all, we would like to thank you for your time and care in reviewing our manuscript, as well as for all your valuable comments. We have tried to implement all them for improving the quality and the readability of the paper.

The structure of the paper is the same of the original submission, as all of you approved it. The main changes have been made in the introduction section and in the methodology (i.e., Material and Methods section): as requested, more contents were included, as well as additional technical information on the proposed system design and the architecture. Moreover, the paper was subjected to an accurate revision of the English language, as recommended.

Below you can find punctual detailed replies for each comment (in red), while in the main text changes related to the contents are yellow highlighted so as to be easily identified.

We hope that you will find the new version more suitable for publication.

Sincerely,

The authors

REVIEWER #3

The author introduced about Low-cost UWB-based Real Time Locating System. The novelty and the proposed work are satisfied with the following comments.

# In the abstract, highlight the novelty by adding a few more features of the proposed model

In the abstract, two brief sentences yellow highlighted were added, so as to comply with your request.

# In the introduction, suggested including the

- key points of node localization of wireless networks with respect to the proposed system

- Add a few more concepts of industrial wireless networks with respect to the proposed model

- Include a few more concepts related to Industrial WSN with RTLS 

refer to the links if needed:

https://www.hindawi.com/journals/wcmc/2022/5290028/

https://www.sciencedirect.com/science/article/pii/S2214785321017582

The introduction was rearranged and new contents were added according to your note, including some of the papers you kindly suggested to us, which were very interesting and useful. Thank you for this tip!

# In Section 2, from 2.1.1 to 2.1.4 =>add the key points and highlight the importance with respect to the proposed model in order to match the following terms Static accuracy, (Dynamic accuracy), Sensitivity, and Start-up time.

The key points of the experiments have been placed at the beginning of the section, pointing out the relevance of the experiment and the performance assessment with specific respect to the considered industrial use case. We hope that in this new version is more suitable than the previous one.

The results with a graphical analysis of various parameters are satisfying the proposed idea.  

#In section 3, Figure 4 - clarity is not satisfied. Change with better quality.

The figure in question (and the following one as well) have been improved by increasing the resolution of the original images; moreover, to the sake of clarity, a second view of the error on a cartesian map was also added (they are the current Figures 6 and 7).

#In section 4, suggested including some additional key points and highlighting the results by comparing the existing work from test 1 to test 6.

With reference to this comment, we agree with you on the fact that it would be interesting to compare our results with existing works. However, to the best of our knowledge, there are no previous similar studies which could be compared with our works. We also carried out a query on the main scientific database (i.e., Scopus and WoS) in order to be further sure on that, and we confirm that no practical industrial tests were found.

# In Conclusion, add the extension of future work by highlighting the limitations of the proposed work

The conclusions have been amended according to the request, including future research; changes are yellow highlighted within the text. We hope that now the paper is completed and suitable for publication on Sensors journal.

Round 2

Reviewer 2 Report

Accepted